# Seasonal and social factors associated with spacing in a wild territorial electric fish

Lucía Zubizarreta[1,2], Laura Quintana[2], Daniel Hernández[3], Franco Teixeira de Mello[4], Mariana Meerhoff[4,5], Renato Massaaki Honji [6], Renata Guimarães Moreira[7], Ana Silva [2,8]*

**1** Laboratorio de Neurofisiología Celular y Sináptica, Departamento de Fisiología, Facultad de Medicina, Universidad de la República, Montevideo, Uruguay, **2** Unidad Bases Neurales de la Conducta, Departamento de Neurofisiología Celular y Molecular, Instituto de Investigaciones Biológicas Clemente Estable, Montevideo, Uruguay, **3** Laboratorio de Control Ambiental, Polo Educativo Tecnológico Arrayanes, CETP-UTU, Maldonado, Uruguay, **4** Departamento de Ecología y Gestión Ambiental, Centro Universitario Regional del Este, Universidad de la República, Maldonado, Uruguay, **5** Department of BioScience, Aarhus University, Silkeborg, Denmark, **6** Centro de Biologia Marinha, Rodovia Manoel Hipólito do Rego, Universidade de São Paulo (CEBIMar/USP), São Sebastião, SP, Brazil, **7** Departamento de Fisiologia, Instituto de Biociências, Universidade de São Paulo, Rua do Matão, São Paulo, SP, Brazil, **8** Laboratorio de Neurociencias, Facultad de Ciencias, Universidad de la República, Montevideo, Uruguay

* asilva@fcien.edu.uy

**Data Availability Statement:** Raw data are available from the https://osf.io/dashboard database Seasonal and social factors associated

## Abstract

In this study, we focused on the seasonal variation of the determinants of territory size in the weakly electric fish *Gymnotus omarorum*. This species is a seasonal breeder that displays year-round territorial aggression. Female and male dyads exhibit indistinguishable non-breeding territorial agonistic behavior and body size is the only significant predictor of contest outcome. We conducted field surveys across seasons that included the identification of individual location, measurements of water physico-chemical variables, characterization of individual morphometric and physiological traits, and their correlation to spatial distribution. *G. omarorum* tolerates a wide range of dissolved oxygen concentration, and territory size correlated positively with dissolved oxygen in both seasons. In the non-breeding season, territory size was sexually monomorphic and correlated only with body size. In the breeding season, territory size no longer correlated with body size but differed between sexes: (i) the overall spatial arrangement was sexually biased, (ii) territory size depended on gonadal hormones in both sexes, which was expected for males, but not previously reported in females, (iii) female territory size showed a positive relationship with gonadal size, and (iv) females showed relatively larger territories than males. This study demonstrates seasonal changes in the determinants of territory size and thus contributes to the understanding of the mechanisms underlying the behavioral plasticity natural territorial behavior.

## Introduction

The mechanisms underlying behavioral plasticity, by which animals respond to dynamic environmental and social contexts, are not fully understood [1]. The study of the modulation of territorial behavior in wild species is especially suited for this aim. In territorial behavior, animals first detect the environmental and social clues that determine territory quality, then

with spacing in a wild territorial electric fish (DOI 10.17605/OSF.IO/65D8A).

**Funding:** This research was funded by Agencia Nacional de Investigación e Innovación (ANII), Fondo Clemente Estable grants to AS (ANII_FCE_6180; FCE_4272) and LQ (ANII_FCE_136381). Programa de Desarrollo de las Ciencias Básicas gave support to LZ, MM, FTM, LQ and AS and Universidad de la República gave support to AS, MM and FTM.

**Competing interests:** The authors have declared that no competing interests exist.

compare this information to their individual requirements and fighting abilities, and finally decide whether to compete over an area. Therefore, the distribution of territorial animals in space provides insights into how animals integrate individual traits with environmental and social factors. Variation in the ability or motivation to obtain and defend a territory can generate differences in territory size because traits such as body mass, sex, and reproductive state are known to influence resource holding potential and resource value [2–6]. Within a population, body size is associated with territory size in most species, as it directly determines metabolic requirements. Moreover, body size strongly impacts contest outcome and territory size [7–10]. In species that display territoriality in both sexes, asymmetries in fighting abilities and motivational factors may cause sex differences in territory size. For example, in red squirrels (*Sciurus vulgaris*), males often hold larger territories than females although this species is sexually monomorphic in body size [11]. In stripped plateau lizards (*Sceloporus virgatus*), females are more territorial than males for motivational reasons [12].

Many species show territorial behavior only during the breeding season [13]. On the other hand, some species show robust territorial aggression all year round even though they are seasonal breeders. Non-breeding territoriality is found in diverse taxa, including birds [14–17], mammals [18–20], reptiles [21], and fish [22,23]. These species offer a valuable opportunity to study the seasonality of environmental features and individual traits, and their relation to territory size in the natural habitat. During the breeding season, male territorial aggression depends largely on gonadal steroids across vertebrates [24–26] and, in particular, androgen levels have been related to territory size in the wild [27–30]. By contrast, in breeding females, there are few studies on the association between circulating estrogen ($E_2$) levels and territorial aggression in free-living conditions [31–33], and, to our knowledge, there are no studies reporting the association between circulating $E_2$ and territory size.

The South American weakly electric fish, *Gymnotus omarorum* [34], is a seasonal breeder that displays male and female territorial aggression all year and thus is an interesting model system to study the seasonal control of territoriality and its sex differences [35]. Previous laboratory results showed that this species presents a remarkably robust non-breeding territorial aggression (described initially in [22]). In these well-characterized agonistic behavioral displays, fish modulate their electric organ discharge (EOD) to signal a submissive [36–38] or dominant status that persists for at least 36 hours [39]. Under experimental laboratory conditions, male-male [40] and female-female [41] dyads that display non-breeding territorial behavior have no differences in contest outcome, temporal dynamics of the agonistic encounter, levels of aggression, or submissive signaling [35]. Moreover, the only significant predictor of contest outcome is body size [22], and none of the features of agonistic encounters depend on circulating gonadal hormones [40].

In this study, we evaluated seasonal variations in the ecological, morphometric, and physiological correlates of territory size in free-living *Gymnotus omarorum*. Based on previous results, we predicted that, in the non-breeding season when gonads are regressed and circulating gonadal hormones are low, territory size would be sexually monomorphic and explained mostly by body size. For fish in the breeding season, when motivational aspects of territoriality may be confounded with the reproductive drive, we expected territory size and/or its determinants to be sexually dimorphic.

## Materials and methods

### Study location and sampling seasons

Fieldwork was performed in the Laguna de los Cisnes, Uruguay (205 ha, 34° 48′ S, 55° 18′ W), which consists of a three-part interconnected shallow system of freshwater (maximum depth 5

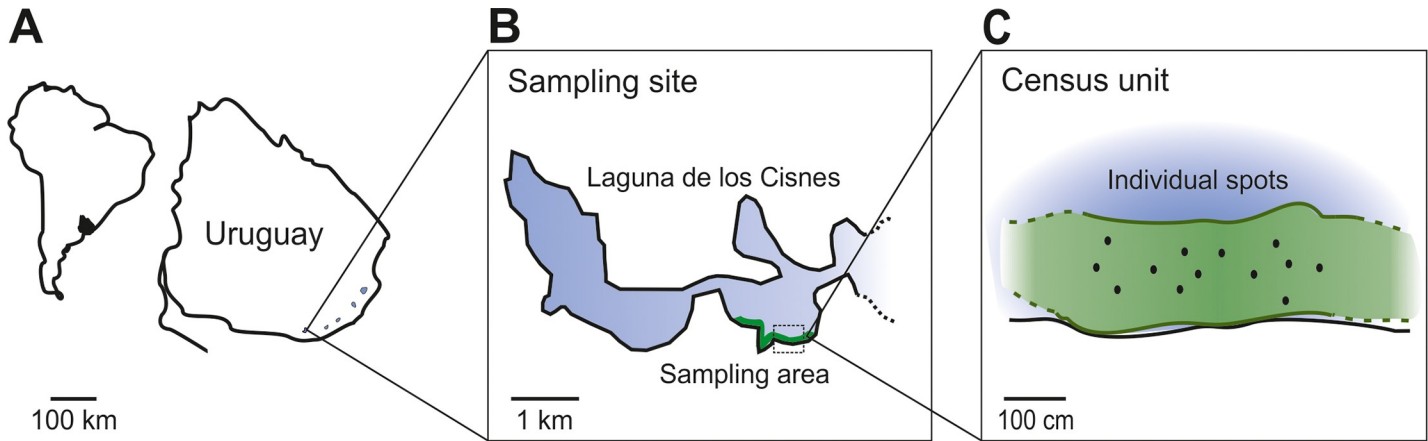

**Fig 1. Study site and sampling method.** A. The study site is located in Maldonado, Uruguay, in Laguna de los Cisnes. B. The shores of the lake have water hyacinths creating extensive floating mats that constitute the sampling area. C. Census unit illustrating individual spots. Fish location in individual spots was achieved by carrying out an electric census (Survey 1). Once a fish was located, water dissolved $O_2$ concentration, temperature, and fish EOD rate were measured in each spot. In Survey 2, individual spots were revisited in sequence, fish were collected, and individual traits measured.

m) with no inputs of salt or brackish water (Fig 1A, [42]). The study species, *Gymnotus omarorum*, is the sole species of weakly electric fish present in the study area. The littoral area of the lake is blanketed by a dense strip (5–40 m width) of free-floating aquatic macrophytes that cover the sampling area (Fig 1B), with *G. omarorum* typically resting among the roots of these plants [34].

Following Quintana et al. (2004), who identified the breeding season for weakly electric fish species in this region from December to February (austral summer), we collected field data during December, corresponding to the early-breeding season before the appearance of offspring [43], and from June to August (austral winter), corresponding to the non-breeding season.

## Sampling

Sampling area was homogeneous in depth, distance to shore, and vegetation composition. The sampling area was divided into adjacent areas referred to as census units (Fig 1B and 1C, defined below), which were each studied on different days, without repeating sites. Sampling was performed during the day, which is the resting phase in this species [44], in two periods: the first one (Survey 1) during the morning (0800-1200h), and the second one (Survey 2) in the afternoon (1300-1800h).

All research procedures complied with ASAP/ABS Guidelines for the Use of Animals in Research and were approved by the Institutional Ethical Committee (Facultad de Ciencias, Universidad de la República, Comisión Honoraria de Experimentación Animal, Exp: 240011-002308-14).

## Field survey 1: Electric census and environmental variables

To achieve an initial estimate of the spatial arrangement of individuals, we performed an electric census during the resting phase of the animals and measured the distance between each fish and its nearest neighbor (distance to the nearest neighbor, DNN) in the two seasons. To calculate DNN, we considered each focal fish distance to the nearest neighbor. DNN was used as a proxy of territory size. Although this is a simplified approach, animals which present territorial aggression display a decrease in defense as distance increases from center points (10,

reviewed in [45]). This study was carried out during the resting phase of *G. omarorum* and fish have been shown to display site fidelity, which suggests they are located in the central parts of their territory. We performed the electric census by locating individual fish by means of an audio amplifier connected to a pair of electrodes, as described elsewhere [46] and individual fish were localized by their EODs when the sound is maximal (detection range is 60 cm). Monitoring the EOD rate allowed us to confirm that fish remain undisturbed during the measurement of parameters since EOD rate increases transiently when fish are alerted by novel stimuli (reviewed in [47]). To perform the electric census, two experienced researchers waded up to 1.2 m depth in the water, to slowly and carefully access areas 1 m away from individual fish. Once a fish was located in an individual spot, the plant above it was tagged (Fig 1C). Water conductivity, dissolved oxygen concentration ($O_2$, mg/l) and temperature (T,˚C) were measured 30 cm below the surface (using TDSu Testr 3, Cole Parmer for conductivity, and Oxy-Ward, Handy Polaris for $O_2$ and T). After taking physico-chemical measurements, the EOD of each fish was recorded *in situ* for 10 s through two electrodes lowered into the vicinity of the animal and connected to an amplifier located on the shore of the lake (World Precision Instruments Inc., Sarasota, FL. DAM-50, AC-coupled). After amplification, EOD signals were recorded on a portable computer, captured by the audio card, and stored for further analysis. To normalize the potential effect of water temperature on EOD rate, values were corrected to a constant 20˚C using the Q10 value of 1.5 as calculated for electric fish, from the equation: Q10 = EOD rate x T / EOD rate x (T + 10) [48,49].

During each sampling day, the measurement of physico-chemical water parameters and EOD rate was conducted for all the fish located within a census unit. A census unit was defined as the area with all fish detected by 12 AM, or the area where we detected a group of fish surrounded by at least 6 m of water uninhabited by the species (Fig 1C). Field survey 1 was performed both, during the breeding and non-breeding seasons.

### Field survey 2: Quantification of individual traits

In addition to environmental variables, individual morphological and physiological traits can influence spatial distribution and vary seasonally. Therefore, based on Survey 1, we characterized individual traits of captured fish in both seasons, and then analyzed their correlation with spatial distribution. Individual spots were revisited in sequence, and each fish located under the tagged plants was collected using a net, without disturbing nearby tagged sites. Netting desorganize plants, thus if the first netting was unsuccessful no other attempts were made in that site and fish was reported as "not recovered". Fish were weighed, measured for length, sacrificed by quick decapitation and their gonads were visually inspected for sex determination. In the breeding season, fish were anesthetized immediately after netting by immersion in a fast-acting eugenol solution (1.2 mg/l, first dissolved in alcohol 70%) and blood was collected from the caudal vein with a heparinized syringe within 3 min of capture, which is the time range usually used to avoid a stress response due to manipulation [50–52]. Blood was placed in tubes on ice in the field and later centrifuged (3000rpm, 10 min) in the laboratory and stored at -80˚C. Dissected gonads were stored in dry ice in the field, and then weighed in the laboratory for gonadosomatic index (GSI) calculation ([Gonad Weight / Total Tissue Weight] x 100 [53]).

### Hormone assays

17-β Estradiol ($E_2$) levels were quantified in breeding females, and 11-Ketotestosterone (11-KT) in breeding males by enzyme-linked immunosorbent assay (ELISA) using commercial kits (IBL International, Hamburg, Germany for $E_2$ and Cayman Chemical Company, MI,

USA for 11-KT). The analyses were performed according to the manufacturer's instructions and a standard curve was run for each ELISA plate. In all cases, samples were assayed in duplicate and analyses were performed on samples whose coefficients of variation were below 20% [54]. Intra-assay variation was 3.95% for $E_2$ (detection limit: 25 pg/ml) and 6.2% for 11-KT (detection limit: 1.56 pg/ml). Pilot assays using three different dilutions of 8 samples (4 samples per sex) were performed to establish the appropriate working dilutions, which were 1:2 for breeding female $E_2$ and 1:30 for breeding male 11-KT. The assays were validated with standards provided in the kit, indicating that each assay effectively detects *G. omarorum* $E_2$ and 11-KT.

## Data analysis and statistics

All data were subjected first to D'Agostino & Pearson normality test. If data fit a Gaussian distribution, they were analyzed with parametric tests; otherwise non-parametric comparisons were used. Parametric, non-parametric statistical analyses, and simple linear regressions were carried out with PAST [55], and generalized linear models (GLMs) and Binomial tests with software R [56] using RStudio interface.

Data analysis was performed in three steps. The first step of analysis used data from Survey 1 to explore environmental heterogeneity. Environmental variables ($O_2$ and T) were compared between seasons by Mann-Whitney *U* test. To analyze $O_2$ and T heterogeneity we calculated the coefficient of variation (CV, SD/mean*100) for each census unit and then the mean CV and standard deviation per season. Finally, we performed a linear regression between two individual traits, DNNs and EOD basal rate, and $O_2$ content, both in the breeding and non-breeding seasons (Fig 2).

The second step of the analysis was based on the dataset obtained in Survey 2, which entailed fish collection. We explored whether individual morphometric and physiological traits correlated with DNN in both the non-breeding and breeding seasons. Individual traits (body size and EOD rate), DNN, and DNN relative to body length were compared between sexes within each season by Student t-tests (Table 1). Because body weight and body length were strongly correlated (R = 0.91, N = 47), and because we had more measurements of body length than of body weight, we used body length as an indicator of body size throughout. Body length, EOD rate, and DNN were compared seasonally by Student t-test for females and males together.

To examine the effects of individual traits (see below) on DNN as the dependent variable, weused GLMs; [57], within each season analyzed separately. For the breeding season, we first ran a model with body length, EOD rate, and sex as explanatory variables. Because the initial model was non-significant, we ran one model for females with body length, EOD rate, and circulating $E_2$ as explanatory variables, and a second model for males with body length and EOD rate as explanatory variables (Table 2). For the non-breeding season, the model included females and males together, and the explanatory variables were body length, EOD rate, and sex (Table 3). For each season separately, initial models contained all single effects and pairwise interactions of the explanatory variables.

To select the most parsimonious GLM, we used the command bestglm [58] for a maximum of three simultaneous variables, and considered up to second order interactions. Initial models were simplified by the stepwise deletion of the least significant terms in a model and compared successive steps of model simplification by the Akaike information criterion (AIC), deleting a term whenever there was a difference of more than two units between alternative models until arriving to the most parsimonious model that could be fitted. The selection of the best model included the AIC criterion as well as the number and statistical significance of the estimated

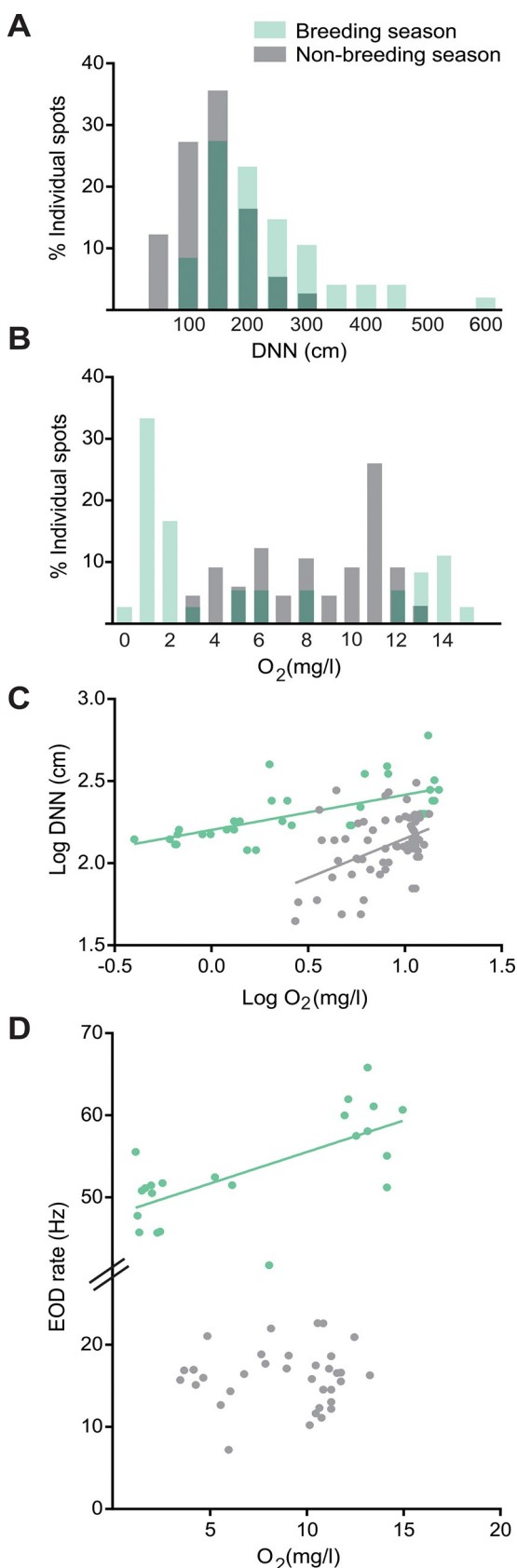

**Fig 2. Fish spatial distribution based on environmental variables (Survey 1).** The breeding season is represented in green and the non-breeding season in gray; dark green sections implies overlap of both seasons. A. Frequency distribution versus DNN(in cm). B. Frequency distribution versus $O_2$ concentration (in mg/l) measured at 30 cm from the surface in each individual spot. C. Linear regression of DNN and $O_2$ concentration in individual spots (Log transformed). Breeding season: $R^2 = 0.44$, p = 1 exp -4, N = 35; non-breeding season: $R^2 = 0.21$, p = 1 exp-4, N = 50. D. Linear regression of EOD rate and water oxygen concentration in individual spots. Breeding season: $R^2 = 0.47$, p = 5 exp-4, N = 22; non-breeding season: $R^2 = 1$ exp-3, p = 0.54, N = 39.

parameters, discarding models with improvements in the AIC but non-significant parameters. All models were subjected to the customary residual analysis [59].

The final step of analysis aimed to address three points. The first point was to further evaluate how sexually dimorphic traits, such as gonad size and sexual hormones, impacted on DNN particularly during breeding (Fig 3). To do this, we compared DNN relative to body size between non-breeding and breeding season separately for each sex. In addition, to analyze the relative importance of gonadal mass on individual spacing, we performed linear regressions of DNN versus GSI, and body weight–gonad weight, both in breeding females and males. The second point was to analyze whether oxygen concentration, besides affecting DNN, showed association with morphological (GSI) and physiological individual traits (EOD rate and $E_2$ levels). The third point was to evaluate whether the sex of the nearest neighbor differed from an expected random distribution, and for that we used Binomial tests [60]. For the analysis, the focal fish was only considered if the nearest neighbor was captured and its sex confirmed. The proportion of sexes expected for a random distribution was deduced from the empirical sex ratio observed in each season (Fig 4).

## Results

In both seasons, individual *Gymnotus omarorum* were typically located in water depths of ~ 30 cm among the dense roots of extensive floating mats of vegetation along the littoral area across seasons (Fig 1B; i.e., in the breeding season during the austral summer, and in the non-breeding season during the austral winter). The free-floating water hyacinth *Eichhornia crassipes* dominated both surface and underwater areas accounting for 86% of the total subaquatic biomass. Associated vegetation was composed of submerged *Egeria densa* and *Miriophyllum aquaticum*, free-floating *Salvinia auriculata*, and the rooted but partly emergent *Ludwigia elegans*, and *Hydrocotyle criptocarpa*. This vegetation, with the same composition, was present all year, although overall coverage was lower during the non-breeding season.

As expected in the subtropical region, water temperature and oxygen concentration showed significant differences across seasons. Water temperature was higher during the breeding than during the non-breeding season ($27.3 \pm 0.1°C$, N = 36 vs $11.3 \pm 0.5°C$, N = 60; p = 1 exp-4,

**Table 1. Sexual comparison of individual traits and DNN in the breeding season and in the non-breeding season (Survey 2).** Individual traits: Body length and EOD rate corrected by water temperature. Distance to the nearest neighbor (DNN in cm), and relative DNN (DNN/body length). Values are expressed as mean ± standard error of the mean (SEM), and statistical comparisons were performed by t-test.

| | Body size (cm) | | EOD rate $_{temp}$ | | DNN (cm) | | DNN / body length | |
|---|---|---|---|---|---|---|---|---|
| | ♀ | ♂ | ♀ | ♂ | ♀ | ♂ | ♀ | ♂ |
| **A- Breeding season** | 22.4 ± 0.5 n = 13 | 24.5 ± 0.9 n = 15 | 41.0 ± 1.1 n = 13 | 39.3 ± 0.9 n = 17 | 220 ± 20 n = 13 | 230 ± 30 n = 18 | 9.8 ± 0.8 n = 13 | 9.9 ± 1.4 n = 17 |
| | ♀ vs ♂ p = 0.2 | | ♀ vs ♂ p = 0.3 | | ♀ vs ♂ p = 0.8 | | ♀ vs ♂ p = 0.9 | |
| **B- Non-breeding season** | 16.9 ± 0.9 n = 26 | 17.4 ± 0.8 n = 27 | 20.0 ± 1.2 n = 18 | 19.3 ± 0.95 n = 18 | 113 ± 10 n = 23 | 120 ± 10 n = 23 | 7.0 ± 0.6 n = 23 | 7.8 ± 0.7 n = 23 |
| | ♀ vs ♂ p = 0.7 | | ♀ vs ♂ p = 0.6 | | ♀ vs ♂ p = 0.5 | | ♀ vs ♂ p = 0.4 | |

**Table 2. GLM models that presented the best adjustment to explain DNN in females during the breeding season with data obtained in Survey 2.** Model intercept and explanatory variables are expressed as Mean (SD). For each parameter, the value is shown in bold if statistically significant (p<0.05), in italic if marginal (p<0.1), and expressed as NS if non-significant. Model 1 and Model 2 did not present significant differences by the Akaike information criterion (AIC).

| Model | Intercept | Circulating E$_2$ | EOD rate | N | P model | Adjusted R2 | AIC |
|---|---|---|---|---|---|---|---|
| M1 | **452** (141) | **0.14** (0.03) | *-0.51* (2.7) | 13 | 0.02 | 0.44 | 20.24 |
| M2 | **188** (18) | **0.1** (0.04) | NS | 13 | 0.03 | 0.31 | 22.21 |

Mann Whitney *U* test), whereas O$_2$ concentration was significantly lower in the breeding compared to the non-breeding season (5.6 ± 0.9 mg/l, N = 36, vs 8.4 ± 0.4 mg/l, N = 65; p = 4 exp-3, Mann Whitney *U* test). By contrast, water conductivity remained consistent (< 150 μS/cm) throughout the year.

## Field survey 1: Fish spatial distribution and environmental variables

Our first step of analysis aimed to relate environmental data to individual spacing and EOD rates of fish detected by electrical census. Fish occurred under the central area of the floating mats, and absent from the edge near the open water (Fig 1C). In both seasons, fish were distributed evenly, and not aggregated with other conspecifics. The distribution of DNNs was asymmetrical, skewed with a mode at 150 cm (Fig 2A). The DNN mean value was significantly higher in the breeding season than in the non-breeding season (230 ± 15 cm, t-test, T = 5.9, p = 1 exp-4, N = 47 vs 140 ± 7 cm, N = 73).

Water temperature showed low variability among individual spots within each census unit (breeding season mean T CV = 2.0 ± 2.4%, non-breeding season mean T CV = 6.5 ± 5.5%). In contrast, O$_2$ concentration showed high variability within each census unit (breeding season mean O$_2$ CV = 20.4 ± 9.3%, non-breeding season mean O$_2$ CV = 25.9 ± 14.6%). Consequently, we explored whether O$_2$ may contribute to fish spatial patterns. During the breeding season, the distribution of O$_2$ concentration ranged from 0 to 15 mg/l and was positively skewed (Fig 2B, green), with a higher frequency of low values (mode at 1 mg/l). However, during the non-breeding season, O$_2$ distribution was negatively skewed, showing a single mode at 11 mg/l (Fig 2B, gray). Oxygen concentration and DNN showed a positive and significant association both in the breeding ($R^2$ = 0.44, p = 1 exp -4, N = 31, Fig 2C green) and in the non-breeding season ($R^2$ = 0.21, p = 1 exp -4, N = 50, Fig 2C gray). Interestingly, O$_2$ showed a positive linear relationship with individual EOD rate during the breeding but not the non-breeding season (breeding: $R^2$ = 0.47, p = 5 exp -4, N = 22, Fig 2D green; non-breeding: $R^2$ = 1 exp -3, p = 0.54, N = 39, Fig 2D gray).

## Field survey 2: Fish spatial distribution based on individual traits

The second step of analysis explored the impact of individual morphological and physiological characteristics of fish on DNN in both the non-breeding and breeding seasons. Across census units, 71 ± 5.8% of fish located under tagged sites in Survey 1 were recovered in Survey 2. There were no sexual differences in body length, EOD rate, DNN, nor DNN relative to body length, in both the breeding and the non-breeding seasons (Table 1). During breeding, females showed mean circulating E$_2$ levels of 293.7 ± 97.7 pg/ml and males had mean circulating

**Table 3. GLM model that presented the best adjustment to explain the distance to the nearest neighbor (DNN) in both sexes during the non-breeding season with data obtained in Survey 2.** Model intercept and explanatory variable are expressed as Mean (SD). For each parameter the value is shown in bold if statistically significant (p<0.05).

| Intercept | Body size | N | P model | Adjusted R2 | AIC |
|---|---|---|---|---|---|
| 36.4 (34) | **5.2** (2) | 48 | 0.01 | 0.11 | 79.75 |

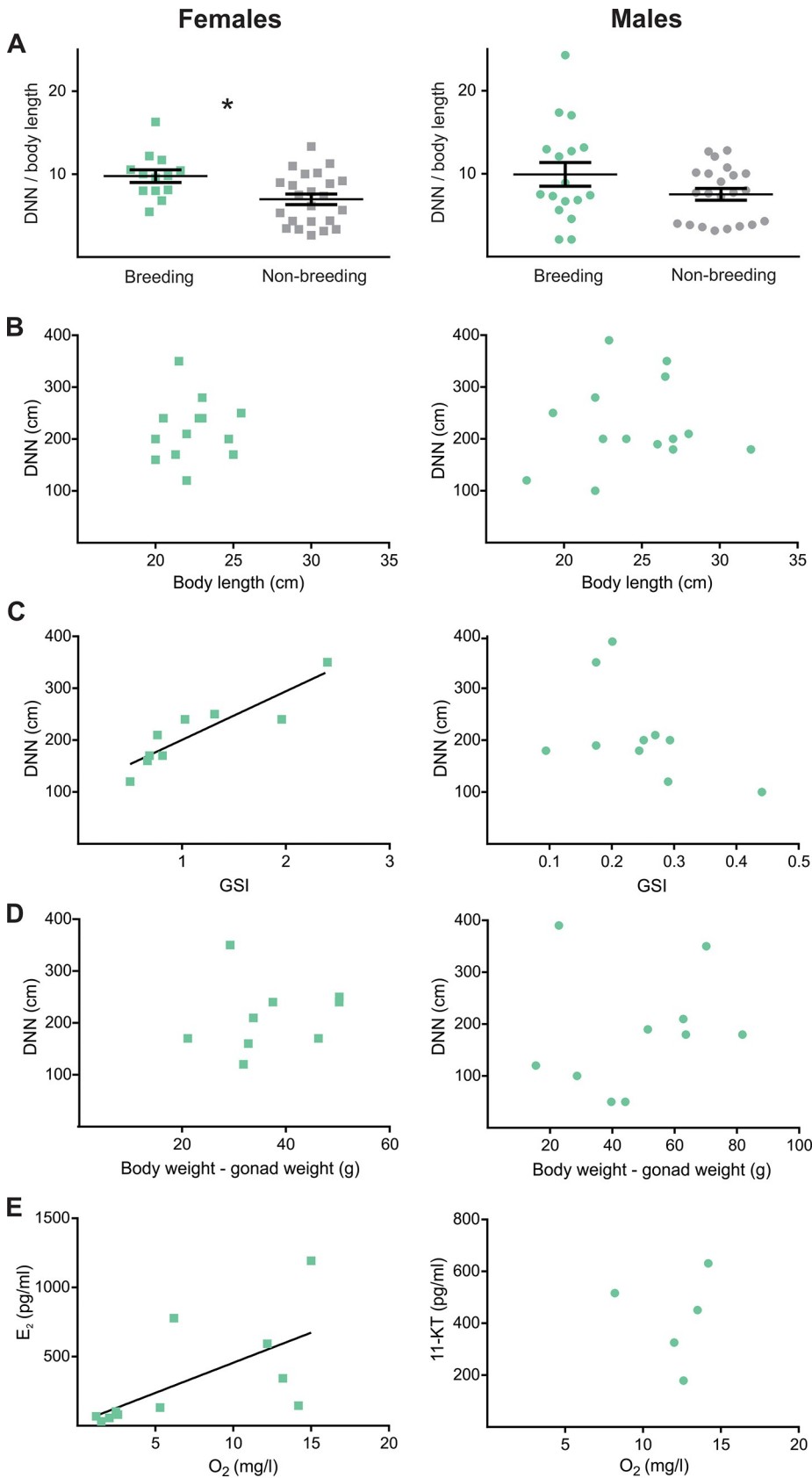

**Fig 3. Emergence of sex dimorphism during the breeding season.** The plots show values for females (left panels, represented as squares) and males (right panels, represented as circles), from data obtained in Survey 2 (Step 3 of analysis). **A-** Relative DNN (cm / cm body length). Dots represent individual values, and in A and B horizontal lines represent mean values, and error bars represent SEM. For each sex, breeding values are shown in the left (green) and non-breeding values in the right (grey). * indicate statistically significance (p < 0.05) t-test. **B-** Linear regression between body size (cm) and DNN (cm) in the breeding season. Females: p = 0.8, $R^2$ = 7 exp-3, N = 13; males p = 0.7, $R^2$ = 0.01, N = 15. **C-** Linear regression between GSI and DNN (cm) in the breeding season. Females: p = 1 exp-3, $R^2$ = 0.8, N = 9; males: p = 0.14, $R^2$ = 0.25, N = 10. **D-** Linear regression between body weight–gonad weight (g) and DNN (cm) in the breeding season. Females: p = 0.7, R2 = 0.02, N = 9; males: p = 0.6, $R^2$ = 0.03, N = 10. **E-** Linear regression between water $O_2$ concentration (mg/l) and $E_2$ (pg/ml) in breeding females: p = 0.03, $R^2$ = 0.41, N = 11. The same relation is shown for 11-KT in breeding males with no statistical test due to small sample size (N = 5).

11-KT levels of 399.1 ± 140.9 pg/ml. Mean adult body size was significantly larger in the breeding season than in the non-breeding season (t-test, T = 6.6, p = 1 exp-4, N breeding = 28, N non-breeding = 53). In addition, EOD rate was higher (t-test, T = 18.8, p = 1 exp-4, N breeding = 30, N non-breeding = 36), DNN was larger (t-test, T = 5.6, p = 1 exp-4, N breeding = 31, N non-breeding = 46) and relative DNN (DNN/body length) was also larger during the breeding season (t-test, T = 2.7, p = 9 exp-3, N breeding = 30, N non-breeding = 46).

We evaluated the determinants of DNN separately in the breeding and non-breeding seasons. Because the first GLM including breeding females and males, with body length, EOD rate, and sex as explanatory variables was not significant, we separated sexes into two different models and included circulating $E_2$ levels as an explanatory variable for females. We obtained two significant models in females, both equivalent according to the AIC criterion (Table 2). In the model with the best adjustment (model 1), DNN showed a positive correlation with circulating $E_2$, and a marginal negative correlation with EOD rate. In model 2, EOD rate was not a significant explanatory variable for DNN, and only $E_2$ had a significant positive correlation.

For males, we found no correlation between the independent variables tested and DNN. Although androgen levels were not included in the model (due to a low number of valid samples), it is worth mentioning that circulating 11-KT levels showed a positive trend with DNN, although the dataset was too small for sufficiently powerful statistical analysis. In the non-breeding season, individual traits were the same for males and females, thus we were able to include both sexes in a single GLM to test the influence of individual traits on DNN. We explored if individual sex, body size, and EOD rate correlated with DNN, and found that body size, but not sex nor EOD rate, correlated positively with DNN (Table 3).

To assess sexual differences in territorial features during breeding, we performed a third step of analysis. We compared relative territory size between seasons and evaluated the relationship between individual and environmental traits. Although both absolute and relative DNN were larger in the breeding season in both sexes, an analysis in which males and females were separated showed that this seasonal difference is only significant in females (Females: breeding 9.8 ± 0.8, N = 13 vs non-breeding 7.0 ± 0.6, N = 23; t-test, T = 2.7, p = 0.01, Males: breeding 9.9 ± 1.4, N = 17 vs non-breeding 7.5 ± 0.7, N = 22; t-test, T = 1.4, p = 0.12, Fig 3A). In accordance with the breeding GLM results, body size showed no association with DNN in the breeding season in either sex (Females: p = 0.8, $R^2$ = 7 exp-3, N = 13. Males: p = 0.7, $R^2$ = 0.01, N = 15; Fig 3B). With respect to the relative importance of gonadal maturity on spacing, we found that GSI was significantly higher in the breeding than in the non-breeding season in females (Females: 1.1 ± 0.22% N = 9 vs 0.6 ± 0.09% N = 11; t-test, T = 2.3, p = 0.02,. Males: 0.24 ± 0.03% N = 10 vs 0.23 ± 0.03, N = 16; t-test, T = 0.3, p = 0.87); and as expected GSI showed a positive relationship with $E_2$ in breeding females (p = 2 exp-3, $R^2$ = 0.76, N = 9, data not shown). In addition, GSI showed a positive relationship with DNN only in breeding females (Females: p = 1 exp-3, $R^2$ = 0.8, N = 9. Males: p = 0.14, $R^2$ = 0.25, N = 10, Fig 3C); and DNN showed no relation with body weight after the subtraction of gonad weight in breeding

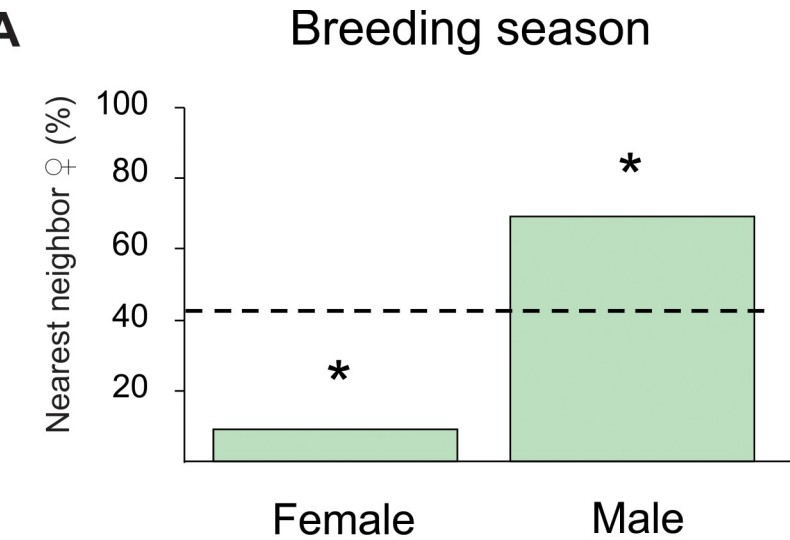

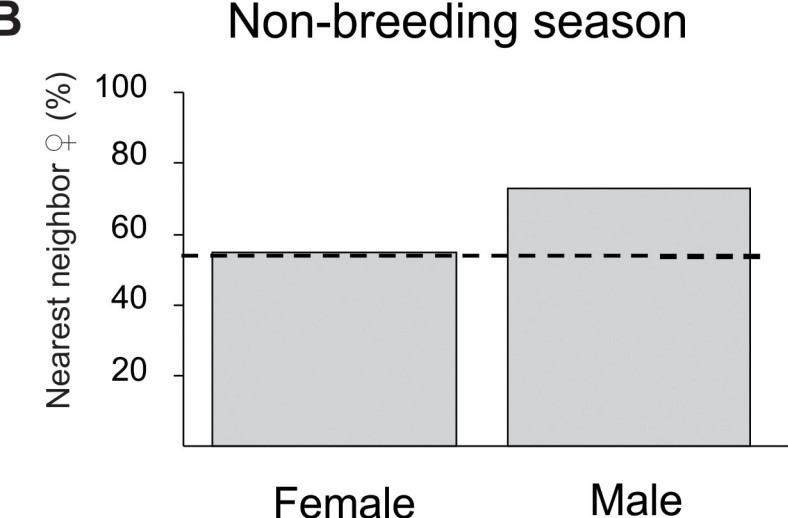

**Fig 4. Seasonality in the spatial distribution of sexes (from data obtained in Survey 2).** Sex of the nearest neighbor (expressed in percentage) when the focal fish is a female or a male, both in the breeding season (A, top), and in the non-breeding season (B, bottom). Dashed line represents the expected percentage to have a female as nearest neighbor for a random distribution, according to the empirical sex ratio (41% in the breeding season, and 50.9% in the non-breeding season). * indicates statistical significance ($p < 0.05$) according to the binomial exact test.

(Females: $p = 0.7$, $R^2 = 0.02$, $N = 9$. Males: $p = 0.6$, $R^2 = 0.03$, $N = 10$; Fig 3D). Further, water $O_2$ content showed a positive relationship with circulating $E_2$ in breeding females ($p = 0.03$, $R^2 = 0.41$, $N = 11$, Fig 3E) and $O_2$ showed a marginal relationship with female GSI ($p = 0.08$, $R^2 = 0.5$, $N = 7$, data not shown); while male 11-KT showed no association with $O_2$, although the dataset was too small for statistical analysis ($N = 5$, Fig 3E).

In the breeding season, fish were distributed non-randomly according to sex. The percentage of females with a female as the nearest neighbor was significantly lower than the random distribution ($p = 0.03$, $N = 11$; Binomial exact test, Fig 4A), and the percentage of females with a

male as the nearest neighbor was significantly higher than random (p = 0.04, N = 16; Binomial exact test, Fig 4A). By contrast, in the non-breeding season, the spatial configuration of the population was random with respect to sex. The probability of having a female as the nearest neighbor did not differ significantly from the random distribution for both females (p = 0.82, N = 20; Binomial exact test, Fig 4B) and males (p = 0.12, N = 15; Binomial exact test, Fig 4B).

## Discussion

This study examined seasonal and sexual differences in the determinants of territory size in a teleost species. First, we found that *Gymnotus omaroum* has a spatial distribution in the natural habitat consistent with territoriality across seasons and that $O_2$ concentration had a significant positive correlation with territory size both in the breeding and in the non-breeding season. In addition, we found that: a) In the non-breeding season, territory size was sexually monomorphic and determined partially by individual body size; and b) there was a seasonal emergence of sexual dimorphism in relative territory size and in its determinants.

In many species, body mass is strongly associated with a variety of physiological and ecological attributes (e.g., home range and metabolic rate) and is key to understanding how animals use the environment [61–64]. The distribution of differently sized animals can be shaped by behavioral interactions such as resource defense [65]. These ideas, put forth initially to analyze interspecific interactions [66], can be also applied to interpret distribution of individuals of the same species, as in this study. We found that body size predicted DNN in the non-breeding season but not in the breeding season (Tables 2 and 3). This suggests that, during reproduction, other physiological, behavioral, and motivational aspects may override body size in predicting DNN, a proxy for territory size.

Oxygen is a limiting physico-chemical variable in aquatic ecosystems, and hypoxia has been related to fish mortality [67]. Interestingly, *Gymnotus* has been shown to tolerate a wide range of dissolved $O_2$ concentrations and can survive in hypoxic environments [68]. *Gymnotus* can breathe air at the water surface and has metabolic adaptations that compensate for any potential damages hypoxia may cause [69]. In our study, individual locations showed $O_2$ concentration correlated with territory size in both seasons (Fig 2C). This suggests that higher levels of dissolved $O_2$ may enable fish to defend large territories because their capacity for aerobic respiration is enhanced. Interestingly, $O_2$ concentration correlated positively with EOD rate only during the breeding season, when EOD rate was significantly higher and $O_2$ content was significantly lower (Fig 2). This is consistent with the metabolic requirement of EOD production [70], which can be specially demanding when EOD rate is high, in the challenging environment of summer hypoxia.

### Non-breeding spacing: Body size dependent and sex independent

In teleosts, year round territoriality has been examined mostly in populations of coral reef communities (for example *Stegastes fuscus*, [71]). Here, we evaluated the seasonal spatial organization in a subtropical freshwater fish in the field and found that spatial distribution during winter was consistent with territoriality. Why would *Gymnotus* defend non-breeding territories? The acquisition and maintenance of territories are known to be mediated by agonistic behavior [72]. In particular, previous reports of laboratory experiments demonstrated that *Gymnotus omarorum* exhibits non-breeding agonistic behavior that mediates territory access [39] and that body size is the main determinant of the fight outcome [22].

The generation of the EOD in electric fish imposes a high basal metabolic cost [70], which requires an additional constant need to forage. In line with this, we found that larger fish hold larger territories, in both males and females (Table 3). Body size is sexually monomorphic in

this species and behavioral experiments showed that agonistic behavior is non-sex-biased [22,35]. Here, we confirmed that non-breeding males and females hold sexually monomorphic territory sizes in the wild (Table 1), probably to cope with energetic requirements that are not expected to differ by sex during winter.

Electric fish use electric communication signals as behavioral displays. Since the EOD encodes information about body size and physiological state [73–77], a fish may maintain territory boundaries in the wild by remotely assessing the EOD of neighbors. EOD rate signals dominant status when fish are kept in close quarters after conflict resolution, but not when allowed to distance themselves [39]. Thus, we expected no correlation between EOD rate and territory size in the field, and this was confirmed by our data (Table 3). Dominance status based on EOD rate may be needed to reinforce submission when a subordinate individual is unable to escape from the dominant as in confined laboratory conditions. However, under the population densities observed in this study, dominance status is not expected.

## Breeding spacing: The emergence of sexual dimorphism in territory determinants

While competition for reproductive opportunities is usually sexually dimorphic, competition over non-sexual resources is usually equal between males and females [78,79]. Moreover, many species with male and female territoriality are sexually monomorphic in body size and signal traits [80–82]. Consistent with this pattern, we found that during the breeding season both absolute and relative territory sizes were sexually monomorphic, which parallels the lack of sex dimorphism in body size (Table 1). However, during breeding, body size did not correlate with territory size (Fig 3B), suggesting that other factors beyond body size influence habitat resource acquisition and maintenance.

Interestingly, a closer analysis (Step 3) showed that although there were no sex differences in relative territory size in each season (Table 1), they emerged when comparing between seasons. Females seemed to use relatively larger territories in the breeding season compared to the non-breeding season (Fig 3A). This result can be interpreted in the context of energetic cost. Female GSI was higher in the breeding season and correlated positively to circulating $E_2$. Moreover, GSI was also an excellent predictor of female territory size (Fig 3C). This suggests that females may establish relatively larger territories in the breeding season related to foraging needs imposed by ovarian maturation. In line with this, circulating $E_2$ positively correlated with territory size in females (Table 2), a result obtained in both breeding female GLM models. EOD rate, on the other hand, is less reliable in its influence upon territory size, as it presented a marginal correlation to territory in only one GLM model.

Steroid hormones are strong modulators of behavioral plasticity [1] as hormones are used to orchestrate organismal-level physiological and behavioral responses to social cues [83,84]. In females, the fact that both GSI and $E_2$ correlated with territory size, combined with reports in which $E_2$ promotes female aggression [85–87], suggests that ovarian $E_2$ modulates territorial behavior in *G. omarorum*. We hypothesize that $E_2$ secretion integrates female metabolic requirements with the social environment, through the expression of territorial behavior. A closer look at the relationship between water $O_2$ concentration and female individual traits, shows that $O_2$ levels correlate positively with circulating $E_2$ (Fig 3E) and marginally with GSI. These data allow us to further speculate that breeding females compete for territories with high $O_2$, which would allow them to better confront energetically expensive processes. In males, we identified a positive trend between circulating 11-KT and territory size, which, although not statistically significant, may be expected given the well documented relationship between androgens and male territoriality [24–26].

The fact that individual body size was larger in the breeding than in the non-breeding season, most likely depends on the physical growth inherent to *G. omarorum* life history. However, we cannot assume that the non-breeding and breeding population remain demographically unchanged. Although this study was not conceived to track population seasonal changes, differences in fish size between seasons may reflect demographic changes. For example, large mature fish may displace smaller adults not only to exploit valuable territories for their own use, but also to access sites where offspring have higher survival as observed, for example, in the cichlid *Lobochilotes labiatus* [88].

Motivation for territory defense in the breeding season may involve not only the value of the territory *per se* (territory size, $O_2$ concentration) but also the sex of the nearest neighbor. The results presented in Fig 4 suggest that *G. omarorum* can assess territory features and use this information to shift positions seasonally. In the non-breeding season individual fish had a closest neighbor that was random with respect to sex, whereas in the breeding season, it was significantly more likely to have an opposite-sex closest neighbor (Fig 4). This evidence supports the idea that the sex of the nearest neighbor becomes a relevant factor for territory value, but only during the breeding season. This can be considered a good example of behavioral plasticity by which individuals respond differently to the same social stimulus (e.g., sex of the nearest neighbor), depending on variations in their internal state (sexually dimorphic hormones).

## Concluding remarks

*Gymnotus omarorum* offers the opportunity to analyze seasonal changes in year-round territoriality. We found no differences in absolute nor relative territory size between sexes across the year. In the non-breeding season, only body size and water $O_2$ concentration influenced territory size. By contrast, in the breeding season, sex became relevant for territorial behavior: a) the sex of neighbors became important; b) territory size was related to gonadal hormones in both sexes, which was expected for males, but not previously reported in females; and c) breeding females used relatively larger territories than non-breeding females, which reflects particular female metabolic demands related to ovarian maturation. This study helps bridge the gap between behavioral plasticity of natural territorial behavior and its underlying mechanisms.

## Acknowledgments

We are very grateful to Adriana Migliaro, Rossana Perrone, Carlos Passos, Federico Reyes, and Bettina Tassino for their useful discussions during the BERTA Workshop, Cerro del Toro, Piriápolis, Uruguay. We also thank the anonymous reviewers for their constructive suggestions that greatly improved this article.

## Author Contributions

**Conceptualization:** Lucía Zubizarreta, Laura Quintana, Mariana Meerhoff, Ana Silva.

**Data curation:** Lucía Zubizarreta.

**Formal analysis:** Lucía Zubizarreta, Daniel Hernández.

**Funding acquisition:** Laura Quintana, Franco Teixeira de Mello, Mariana Meerhoff, Ana Silva.

**Investigation:** Lucía Zubizarreta, Laura Quintana, Ana Silva.

**Methodology:** Lucía Zubizarreta, Laura Quintana, Franco Teixeira de Mello, Mariana Meerhoff, Renato Massaaki Honji, Renata Guimarães Moreira, Ana Silva.

**Project administration:** Laura Quintana, Ana Silva.

**Resources:** Lucía Zubizarreta, Laura Quintana, Mariana Meerhoff, Ana Silva.

**Supervision:** Laura Quintana, Mariana Meerhoff, Ana Silva.

**Visualization:** Lucía Zubizarreta.

**Writing – original draft:** Lucía Zubizarreta, Laura Quintana, Ana Silva.

**Writing – review & editing:** Lucía Zubizarreta, Laura Quintana, Daniel Hernández, Franco Teixeira de Mello, Mariana Meerhoff, Renato Massaaki Honji, Renata Guimarães Moreira, Ana Silva.

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
