## [Decision Letter · Decision Letter 0]

17 Mar 2020

PONE-D-20-02253

Seasonal and social factors associated with spacing in a wild territorial electric fish

PLOS ONE

Dear Dr. Silva,

Thank you for submitting your manuscript to PLOS ONE. After careful consideration, we feel that it has merit but does not fully meet PLOS ONE’s publication criteria as it currently stands. Therefore, we invite you to submit a revised version of the manuscript that addresses the points raised during the review process.

Unfortunately the second reviewer did not return the review after being given extra time to respond. Please respond to the revisions to the one reviewer. I will send the revisions to one additional reviewer at that point in time. Sorry for the delay.

We would appreciate receiving your revised manuscript by May 01 2020 11:59PM. To enhance the reproducibility of your results, we recommend that if applicable you deposit your laboratory protocols in protocols.io, where a protocol can be assigned its own identifier (DOI) such that it can be cited independently in the future. For instructions see: http://journals.plos.org/plosone/s/submission-guidelines#loc-laboratory-protocols

We look forward to receiving your revised manuscript.

Kind regards,

Melissa J. Coleman

Academic Editor

PLOS ONE

Journal Requirements:

3. To comply with PLOS ONE submissions requirements, please provide methods of sacrifice in the Methods section of your manuscript.

Reviewers' comments:

Reviewer's Responses to Questions

**Comments to the Author**

1. Is the manuscript technically sound, and do the data support the conclusions?

Reviewer #1: Yes

2. Has the statistical analysis been performed appropriately and rigorously? 

Reviewer #1: Yes

3. Have the authors made all data underlying the findings in their manuscript fully available?

Reviewer #1: Yes

4. Is the manuscript presented in an intelligible fashion and written in standard English?

Reviewer #1: Yes

5. Review Comments to the Author

Reviewer #1: The authors examine the physio-chemical, hormonal, behavioral and morphological correlates of territory size in a natural population of weakly electric fish. They conclude that territory size varies with oxygen concentration, season, hormonal/gonadal state (in breeding females) and body size (in non-breeding fish). Finally, fish show different affilation seasonally based on sex. This is an impressive data set, one that is rare in teleost fish and unprecedented in electric fish. The data appear carefully collected and statistically analyzed appropriately. The authors test in the field certain relationships that have been thoroughly documented in the laboratory. This paper makes a significant extention of behavioral physiology in the real world of the fish. It will be of great interest to the electric fish community, but also to a broad range of researchers that try to link behavioral physiology with ecology.

Major suggestions:

1. One highlighted conclusion of the paper is that territory size is sexually monomorphic in the non-breeding season, but dimorphic in the breeding season, with females having larger territory sizes than males in the breeding season. The authors speculate that this larger territory size corresponds to a the greater energetic demands of breeding females.

However, this sexual difference only applies when they examine body size-corrected territory size (DNN/body length). Several issues:

a) the authors should be more explicit about why they used body-size corrected territory size for this comparison, but not for other analyses. If it is important to size correct in this comparison, wouldn’t it be important in the general statistical analyses?

b) It is not clear why they corrected for body size using body length rather than body mass. (They collected both these measurements.) Presumably, energetic demands (e.g. metabolic rate) is a function of body mass, not length. They could even analyze the gonad-free body mass as well as the whole body mass to see if part of the apparent seasonal change in territory size is attributable to the relatively large gonads of females during the breeding season. Or even assess whether DNN correlated to absolute size in female to see if there is evidence that the added mass of a large gonad necessitates a larger territory.

c) In looking at the population means for DNN and body length (Table 1), it looks like the dimorphism in size-corrected DNN arises more from the fact that mean female body size changes less (~32%) seasonally than mean male body size changes (~41%)… rather than sexual differences in the seasonal change in DNN. That is, the seasonal change in this ratio is driven more by the denominator than the numerator. In this regard, the seasonal change may be more about demographic changes that behavioral/physiological changes. This, then, begs the question of why mean body size changes seasonally (see below) and why the change in mean body size differs between sexes

2) Because one of the main conclusions (size corrected sex-differences in territorial behavior) depends on seasonal changes in mean body length, the authors should provide some speculation about why this occurs. What happens to the large fish in the non-breeding season? Do they die? emigrate? Conversely, do the large fish drive out the small fish from the study area during the breeding season?

3) The correlation of territory size with 02 concentration is very interesting. Indeed, it is remarkable that fish can be larger in body size, discharge at high rates, respire at high temperature, defend larger territories and grow their gonads--- all very metabolically expensive -- when many of them exist in waters that are <2mg/L O2. But, again, why was DNN not size-corrected, since O2 requirements depend on body mass? Also, why was O2 concentration not simply incorporated as a covariate in the generalized linear model? It would also be interesting to know if EOD rate or GSI (two metabolically expensive traits) correlated with O2 concentration.

4) Finally, it would help to state if there were any correlations based on the position of the territory within the field site. For example, were territories on the edge of the mat of plants different in size or O2 concentration? Was one sex more likely to inhabit “edge” territories?

Minor comments:

Line 144 & 396: The authors synonomize DNN and territory size, given that DNN is a “proxy” for territory size. Please provide a reference that DNN (a linear measurement) is widely recognized as a reliable predictor of territory area. One could easily imagine that a fish might affiliate closely with another fish (small DNN) but still defend a large territory area.

Line 268: I am don’t understand how the population mean of DNN in the non-breeding season could be 1.4m when the mean for males is 1.2 and for females is 1.13 (Table 1). Also the sample sizes in line 268 are different that those in Table 1. Is this a different data set? Also, it would be helpful to put the mean size-corrected DNN here, since it would show the size-corrected DNN changes less seasonally than absolute DNN because mean body size increases seasonally.

Line 304: It would be helpful to include the correlational structure in E2, GSI and EOD rate, given that all three correlate significantly with DNN. Do relatively large gonads indeed predict high E2 levels? Is there experimental evidence in Gymnotus that E2 causally reduces EOD rate?

Line 319 (Table 1) it might also be useful to include raw values of EOD rate, since this would give the reader a sense of how much less active (in the EOD sense) the fish are in the non-breeding season, and thereby require smaller territory for foraging.

6. PLOS authors have the option to publish the peer review history of their article (what does this mean?). If published, this will include your full peer review and any attached files.

Reviewer #1: No

---

## [Author Response · Author response to Decision Letter 0]

6 Apr 2020

Response to Editor and Reviewer comments are all included in the document ''Response to Reviewers'', and some general aspects in the document ''Cover letter''.

---

## [Decision Letter · Decision Letter 1]

8 May 2020

PONE-D-20-02253R1

Seasonal and social factors associated with spacing in a wild territorial electric fish

PLOS ONE

Dear Dr. Silva,

Thank you for submitting your manuscript to PLOS ONE. After careful consideration, we feel that it has merit but does not fully meet PLOS ONE’s publication criteria as it currently stands. Therefore, we invite you to submit a revised version of the manuscript that addresses the points raised during the review process.

Both reviewers had some suggestions for editing the writing. 

We would appreciate receiving your revised manuscript by Jun 22 2020 11:59PM. To enhance the reproducibility of your results, we recommend that if applicable you deposit your laboratory protocols in protocols.io, where a protocol can be assigned its own identifier (DOI) such that it can be cited independently in the future. For instructions see: http://journals.plos.org/plosone/s/submission-guidelines#loc-laboratory-protocols

We look forward to receiving your revised manuscript.

Kind regards,

Melissa J. Coleman

Academic Editor

PLOS ONE

Reviewers' comments:

Reviewer's Responses to Questions

**Comments to the Author**

1. If the authors have adequately addressed your comments raised in a previous round of review and you feel that this manuscript is now acceptable for publication, you may indicate that here to bypass the “Comments to the Author” section, enter your conflict of interest statement in the “Confidential to Editor” section, and submit your "Accept" recommendation.

Reviewer #1: (No Response)

Reviewer #2: All comments have been addressed

2. Is the manuscript technically sound, and do the data support the conclusions?

Reviewer #1: Yes

Reviewer #2: Yes

3. Has the statistical analysis been performed appropriately and rigorously? 

Reviewer #1: Yes

Reviewer #2: Yes

4. Have the authors made all data underlying the findings in their manuscript fully available?

Reviewer #1: Yes

Reviewer #2: Yes

5. Is the manuscript presented in an intelligible fashion and written in standard English?

Reviewer #1: Yes

Reviewer #2: Yes

6. Review Comments to the Author

Reviewer #1: The authors have thoroughly addressed all previous comments. They have done an excellent job reanalyzing portions of their data sets, clarifying their analyses, and revising their manuscript in response.

I have only a few remaining editorial suggestions, listed below.

Line 70: “body size”

Line 80 “By contrast”

Line 110, comma before “which”

Line 164, no need to use parentheses here

Line 165, “through two electrodes lowered into the vicinity of the animal and connected to an amplifier…”

Line 169, probably no need to define Q10 here.

Line 177, “was carried out during both the breeding and non-breeding season”

Line 185, “Netting requires…”

Line 193 “Dissected gonads were …”

Line 218: Data analysis was carried out in three steps.

Line 226: “”… whether individual morphometric and physiological traits correlated with DNN…

Line 230: “Because body weight and length were…”

Line 244: “… and considered up to second…”

Line 247 “two”

Line 263 “… its sex confirmed”

Line 320 “The second step of the analysis explored…”

Line 327 “Mean adult body size was significantly larger in the breeding season than non-breeding season.”

Line 445: Move citation forward. That is “… to analyzed interspecific interactions (66), “

Line 459 “… production (70), which can be….”

Line 496: “… body size influence habitat….” (eliminate ‘on’)

Reviewer #2: (No Response)

7. PLOS authors have the option to publish the peer review history of their article (what does this mean?). If published, this will include your full peer review and any attached files.

Reviewer #1: No

Reviewer #2: No

---

## [Author Response · Author response to Decision Letter 1]

21 May 2020

We addressed Reviewers comments on the file named "Response to reviewers"

---

## [Editor Report · Decision Letter 2]

26 May 2020

Seasonal and social factors associated with spacing in a wild territorial electric fish

PONE-D-20-02253R2

Dear Dr. Silva,

We are pleased to inform you that your manuscript has been judged scientifically suitable for publication and will be formally accepted for publication once it complies with all outstanding technical requirements.

With kind regards,

Melissa J. Coleman

Academic Editor

PLOS ONE
---

## [Editor Report · Acceptance letter]

29 May 2020

PONE-D-20-02253R2 

Seasonal and social factors associated with spacing in a wild territorial electric fish 

Dear Dr. Silva:

I am pleased to inform you that your manuscript has been deemed suitable for publication in PLOS ONE. Congratulations! Your manuscript is now with our production department. 

With kind regards,

on behalf of

Dr. Melissa J. Coleman 

Academic Editor

PLOS ONE